# DNA Interaction with a Polyelectrolyte Monolayer at Solution—Air Interface

**DOI:** 10.3390/polym13162820

**Published:** 2021-08-22

**Authors:** Nikolay S. Chirkov, Richard A. Campbell, Alexander V. Michailov, Petr S. Vlasov, Boris A. Noskov

**Affiliations:** 1Institute of Chemistry, Saint Petersburg State University, 198504 St. Petersburg, Russia; n.chirkov@spbu.ru (N.S.C.); mav030655@gmail.com (A.V.M.); petr_vlasov@mail.ru (P.S.V.); 2Division of Pharmacy and Optometry, Faculty of Biology, Medicine and Health, University of Manchester, Manchester M13 9PT, UK; richard.campbell@manchester.ac.uk

**Keywords:** DNA, adsorption kinetics, poly(*N*,*N*-diallyl-*N*-alkyl-*N*-methylammonium chloride), polyelectrolytes, Langmuir monolayers, dynamic surface tension, dilatational surface rheology, network formation

## Abstract

The formation of ordered 2D nanostructures of double stranded DNA molecules at various interfaces attracts more and more focus in medical and engineering research, but the underlying intermolecular interactions still require elucidation. Recently, it has been revealed that mixtures of DNA with a series of hydrophobic cationic polyelectrolytes including poly(*N*,*N*-diallyl-*N*-hexyl-*N*-methylammonium) chloride (PDAHMAC) form a network of ribbonlike or threadlike aggregates at the solution—air interface. In the present work, we adopt a novel approach to confine the same polyelectrolyte at the solution—air interface by spreading it on a subphase with elevated ionic strength. A suite of techniques–rheology, microscopy, ellipsometry, and spectroscopy–are applied to gain insight into main steps of the adsorption layer formation, which results in non-monotonic kinetic dependencies of various surface properties. A long induction period of the kinetic dependencies after DNA is exposed to the surface film results only if the initial surface pressure corresponds to a quasiplateau region of the compression isotherm of a PDAHMAC monolayer. Despite the different aggregation mechanisms, the micromorphology of the mixed PDAHMAC/DNA does not depend noticeably on the initial surface pressure. The results provide new perspective on nanostructure formation involving nucleic acids building blocks.

## 1. Introduction

The compaction of deoxyribonucleic acid (DNA) and formation of its aggregates in nature has inspired the creation of various DNA nanostructures for medical and technological applications [1,2,3,4]. The use of DNA as a building block of complex nanostructures was first proposed by Seeman in 1982 [5]. Since then, a variety of supramolecular structures of different morphology have been assembled using a technique known as DNA origami [1]. This method, proposed by Rothemund [6], allows obtaining various structures by means of hybridization of single stranded DNA molecules. At the same time, the assembly of double stranded DNA molecules into 2D structures at various interfaces has also been reported [2,7,8,9,10], although in some cases its mechanism remains insufficiently clear.

DNA binding to Langmuir monolayers of lipids and cationic surfactants has been studied for more than twenty years mainly due to a sustained interest in the non-viral gene delivery and construction of DNA-based electronic devices [3,11,12,13,14,15,16,17,18]. It has been shown that DNA conformations in the surface layer largely depend on the properties of the substances used to form a monolayer. For example, Erokhina et al. demonstrated that the interactions of an octadecylamine (ODA) monolayer with oligomers of double stranded DNA in solution can cause DNA splitting [19]. DNA molecules are also known to form ordered domains upon interaction with a cationic lipid monolayer [20,21]. Divalent metal cations facilitate DNA interactions with zwitterionic lipid monolayers leading to a partial penetration of the macromolecules into the monolayer [22,23]. It has also been shown that DNA can form fibrous aggregates at the water-air interface as a result of interactions with cationic surfactants and lipids [24,25,26,27]. The DNA binding to lipid monolayers occurs mainly as a result of hydrophobic and electrostatic interactions [16]. The subsequent transfer to a solid substrate by Langmuir–Blodgett technique after such binding can be employed to fabricate DNA nanostructures of different morphology [28], which find their use in biosensing applications [29,30]. However, little is known about the mechanism of complex DNA layer formation, while this information may be vital for the production of devices with the desired properties. Only a few authors have investigated the kinetics of DNA penetration into spread monolayers of amphiphilic substances from an aqueous subphase. Hansda et al. demonstrated that the interaction of DNA with an ODA monolayer is a very slow process and the surface pressure does not reach a constant value even in 8 h after the injection of a DNA solution [31]. Conversely, the results of UV-visible reflectometry indicated that the equilibrium was reached in less than an hour [32,33]. This apparent discrepancy between the results is presumably a consequence of different surface pressures of ODA monolayers in these studies. In the latter case the initial surface pressure of the ODA monolayer was above 10 mN/m “so that the DNA inclusion is favored” [33], compared to zero initial surface pressure in the former case. At the same time, Janich et al. found out that it took at least several hours for the surface pressure of a cationic lipid monolayer, which was spread onto a DNA solution and compressed to 10 mN/m, to reach a constant value [34]. In that work, an unusual minimum of the dynamic surface pressure has also been observed, although not explained.

It is known that dynamic surface rheology can be useful in the investigation of an adsorption mechanism of macromolecules at the liquid surface [35,36], and provide insight into molecular conformations in the surface layer [37]. This technique together with optical methods and probe microscopy have been recently applied to mixed solutions of DNA and cationic surfactants [27]. In this case, the formation of a network of fibrous aggregates at the liquid surface at certain DNA/surfactant ratios was observed. A similar behavior was also reported in our recent work on solutions of the mixture of DNA and synthetic polyelectrolytes [38]. Indeed, while previous works have addressed the conformation of DNA with synthetic polyelectrolytes at the solution–solid interface [39,40], far fewer studies have addressed such interactions at the solution—air interface. Our recent work on the interactions of DNA with synthetic polyelectrolytes revealed that fibrous aggregates formed at the solution surface depend both on the hydrophobicity of the synthetic polyelectrolyte and the bulk molar mixing ratio [38].

In this study, we aim to elucidate the mechanism of the network formation of DNA interacting with an insoluble cationic synthetic polyelectrolyte–poly(*N*,*N*-diallyl-*N*-hexyl-*N*-methylammonium) chloride (PDAHMAC) monolayer at the solution—air interface for the first time, as studied with several complementary rheological and optical techniques. To this aim, the novelty of our approach is first to localize the synthetic polyelectrolyte by spreading it at the interface at a relatively high solution ionic strength, where it forms an insoluble layer, before DNA is injected into the subphase bulk.

## 2. Materials and Methods

### 2.1. Materials

Calf thymus DNA was purchased from Sigma-Aldrich and used as received. PDAHMAC was synthesized as described previously by the radical polymerization of *N*,*N*-diallyl-*N*-hexyl-*N*-methylammonium chloride [38,41]. The monomer (2.225 g, 9.6 mmol), 4,4′-azobis(4-cyanovaleric acid) (84 mg, 0.3 mmol) and deionized water (0.690 g) were vortexed in a glass ampoule until dissolution, the mixture was subjected to three freeze-pump-thaw cycles in order to remove oxygen, sealed and heated at 70 °C for 13 h. The highly viscous product was dissolved in water and dialyzed against deionized water for three days using Spectra/Por MWCO 6-8 000 (Serva, Heidelberg, Germany). The solution was concentrated in vacuum, frozen and lyophilized. White sponge-like product was dried in vacuum over phosphorous pentoxide, yield: 75.1%, intrinsic viscosity: 0.30 dL/g (1 M NaCl, 25 °C). NMR ^1^Н (D_2_O, 400 MHz) δ, ppm: 3.95 (br. m, 2H, Pyrr-2,5 pseudo equatorial), 3.47–3.38 (br. m, 4H, Pyrr-2,5 pseudo axial, Hex(1)) 3.25 & 3.15 (s, 3H, CH_3_N), 2.77 (br. m 1.6H, Pyrr-2,3 cis), 2.33 (br. m 0.4H, Pyrr-2,3 trans), 1.98–1.22 (br. m, 12H, CH_2_-Pyrr, Hex(2-5)), 0.98 (br. t, 3H, Hex(6)). Sodium chloride (Vekton, Russia) was heated in a muffle furnace for 6 h at about 800 °C to eliminate organic impurities. The Trizma base was purchased from Sigma-Aldrich (Germany) and used without further purification. All the solutions were prepared in triply distilled water.

### 2.2. Sample Preparation

DNA fibers were dissolved in 10 mM Tris-HCl buffer solution containing 20 mM NaCl at pH 7.6. The stock DNA solution was stored at 4 °C for no longer than 2 weeks after preparation. The final concentration of DNA varied between 7 × 10^−6^ and 5.6 × 10^−5^ М (in nucleotide units). Volumes of 115, 60 and 25 μL of 0.2 g L^−1^ PDAHMAC solution in methanol (Vekton, Russia) were spread onto the surface of the buffer solution (surface area 1.08 × 10^−2^ m^2^) to reach the surface pressures of 11, 7 and 2 mN/m, respectively. After that, an appropriate amount of the DNA stock solution was carefully injected into the subphase by a Hamilton syringe. All the used glassware and the Teflon Langmuir trough were cleaned by sulfochromic mixture, washed by distilled water and dried before use.

### 2.3. Methods

#### 2.3.1. Dilatational Surface Elasticity and Surface Tension

The dynamic dilatational surface elasticity was measured by the oscillating barrier method [42]. Periodic expansions/compressions of the liquid surface were created by a Teflon barrier moving back and forth along polished brims of a rectangular Langmuir trough. The frequency and amplitude of the oscillations were kept constant and equaled 0.1 Hz and 2.5%, respectively. The induced oscillations of the surface tension were recorded by the Wilhelmy plate method. The rectangular glass plate, sandblasted to ensure complete wetting, was connected to an electronic microbalance.

The real ε_r_ and imaginary ε_i_ components of the dilatational dynamic surface elasticity ε were calculated according to the following relation:$$\varepsilon =\frac{d\gamma }{dlnA}\;=\varepsilon_{r}+i\varepsilon_{i}=\frac{A\delta \gamma cos\varphi }{\delta A}+\frac{iA\delta sin\varphi }{\delta A}$$
where δγ and δA are the increments of the surface tension and surface area, respectively, and φ is the phase shift between the oscillations of these quantities. The experimental errors of the oscillating barrier method are mainly determined by the errors of the surface tension measurements and are less than ±5%.

The solution surface before the monolayer spreading was cleaned by a Pasteur pipette, which was connected to a pump. All the measurements were carried out at 20 ± 1 °C.

#### 2.3.2. Ellipsometry

A null ellipsometer (Multiskop, Optrel GbR, Germany) with a He-Ne laser (632.8 nm), a fixed compensator (45°) and a two-zone averaging nulling scheme was used to estimate relative changes of surface properties. Measurements of the ellipsometric angles were performed at an angle of incidence close to the Brewster angle of water (49°) to ensure the high sensitivity of the instrument. The electric field of elliptically polarized light can be divided into two components parallel and perpendicular to the plane of incidence. The light reflection from the investigated interface changes the amplitude and phase of these two components. The ratio of the two scalar reflection coefficients of the two components ψ and the corresponding change of the difference of two phases Δ after reflection are related to two complex reflection coefficients r_p_ and r_s_ as follows:$$\frac{r_{p}}{r_{s}}=\tan(\psi )ei^{\Delta }$$

The ratio of the reflection coefficients depends on the wavelength of the incident light, the angle of incidence, the refractive indexes of the bulk phases, and the refractive index and thickness of the surface film. The exact relation can be obtained within the framework of a specific model of the system under investigation. In the basic model of a single-component thin isotropic layer of uniform density between two phases, the difference of ellipsometric angles Δ_s_ between those of the investigated system (∆) and of the subphase (Δ_0_) is proportional to the adsorbed amount Γ [43]. For a mixed system, Δ_s_ may simply be taken as a qualitative measure of the extent of adsorption.

An additional feature of ellipsometry is that fluctuations in the signal may be taken as a measure of lateral homogeneity in the surface layer, such as from the presence of surface aggregates [44]. The fluctuations arise when lateral features with a different refractive index profile normal to the solution surface float in and out of the probed laser beam (size ~ 1 mm^2^) with time. Importantly, if the fluctuations are both positive and negative, the lateral features must have a thickness of more than tens of nanometers due to the cyclic nature of the ellipsometric response; if the fluctuations are all positive then this indicates, however, the lateral features must remain in the thin film limit of some nanometers [45].

#### 2.3.3. IRRAS

IRRAS spectra were recorded using Nicolet 8700 FTIR spectrometer (Thermo Scientific, Waltham, MA, USA) equipped with a Tabletop Optical Module (TOM) described in [46]. The registration of IR single beam spectra was performed with a MCT-D detector. The IR beam in the TOM was focused on the surface of a sample using BaF_2_ lens with a focal length of 750 mm. A linear wire grid polarizer mounted on a motorized rotary translator was used to form polarized IR radiation. The polarizer positioning accuracy was about 0.1 degree. Spectrometer and TOM were purged with nitrogen. In all spectra measurements, 1024 scans were accumulated with a resolution of 2 cm^−1^. Data were collected at an angle of incidence equal to 40°. IRRAS data are defined as plots of reflectance-absorbance (RA) versus wavenumber:$$\mathrm{RA}=-\log\frac{R}{R_{b}},$$
where R and R_b_ are the reflectivities from solutions with a spread layer and pure buffer solutions, respectively. Alignment of the equipment was checked according to the procedure described in [46].

#### 2.3.4. Surface Pressure-Area Isotherms

The surface pressure-area isotherms of a PDAHMAC monolayer were recorded by a Langmuir film balance (KSV NIMA, Helsenki, Finland-Sweden). The surface compression/expansion was executed by two Teflon barriers moving in opposite phases along polished brims of a Langmuir trough. The surface pressure was recorded by the Wilhelmy plate method using a paper plate.

#### 2.3.5. Atomic Force Microscopy

The mixed films of DNA and PDAHMAC were transferred from the liquid surface onto a freshly cleaved mica plate by the Langmuir−Schaefer technique [47], and was dried for at least 3 days in a desiccator at 4 °C before the investigation by atomic force microscopy (AFM) using NTEGRA Prima and NTEGRA Spectra setups (NT-MDT, Zelenograd, Russia) in a tapping mode.

## 3. Results

### 3.1. PDAHMAC Layers on the Surface of Buffer Solutions

To the best of our knowledge spread layers of poly(*N*,*N*-diallyl-*N*-alkyl-*N*-methylammonium) chlorides have not been studied previously. The spreading of PDAHMAC onto the surface of pure water does not lead to a layer formation due to the polyelectrolyte dissolution. The solubility decreases with increase of the ionic strength of the subphase, however, and one can observe a monolayer formation on the surface of 10 mM Tris-HCl buffer solution containing 20 mM NaCl at pH 7.6. The surface pressure isotherm has a quasiplateau at surface pressures close to 11 mN/m (Figure 1). This feature is presumably connected with a two-dimensional phase transition and is not observed at the subsequent expansion. Moreover, the surface pressures at expansion shift strongly to lower surface areas. This hysteresis decreases at the second cycle of compression/expansion when the surface pressure at compression starts to deviate from zero at much lower surface areas. The subsequent cycles of compression/expansion result in a gradual decrease of the hysteresis but the quasiplateau region preserves at compression. Another peculiarity of the surface pressure isotherms consists in the increase of the maximum surface pressure with the increase of the number of compressions and expansions. The observed behavior can be explained by the formation of dense PDAHMAC aggregates in the quasiplateau region at compression. These aggregates do not merge at the higher compression, and are preserved even at expansion. Therefore, the surface pressure increases during the next compression step mainly due to the increase of the surface concentration of non-aggregated polyelectrolyte until the quasiplateau, where the number of surface aggregates increases again. The further compression results in the interaction between the aggregates, and the increase of surface pressure. The maximal surface pressure increases with the surface concentration of dense aggregates and thereby with the number of cycles of compression/expansion (Figure 1).

The AFM images of PDAHMAC layers at surface pressures below and above the transitional region corroborate the proposed interpretation of the surface pressure isotherms (Appendix A). Below and at the beginning of the plateau region the PDAHMAC layer contains a large number of small aggregates with a relatively small number of gaps in an almost continuous layer of the synthetic polyelectrolyte (Appendix A). At higher surface pressures beyond the transitional region, the layer contains much larger surface aggregates (islands), which are sometimes connected by narrow “bridges” (Appendix A). At the same time, the PDAHMAC layers at high surface pressures contain a large number of almost empty gaps. The formation of large aggregates is presumably a consequence of hydrophobic interactions between alkyl chains of the polyelectrolyte. In this case the strong electrostatic repulsion between the large aggregates can result in the formation of the gaps between them and ensure the high surface pressure. These surface aggregates are relatively stable and do not dissolve easily during the layer expansion. Thereby the increase of the aggregate surface concentration in the course of the consecutive compressions leads to a narrowing of the quasiplateau region and the hysteresis loop.

### 3.2. DNA Penetration into a PDAHMAC Layer, Plateau Region

The PDAHMAC layer is stable in the plateau region and both the surface pressure and dynamic surface elasticity change only slightly in the course of continuous oscillations for more than 14 h (Figure 2a,b, black squares). Note that the imaginary part of the complex dynamic surface elasticity of the investigated systems proved to be much less than the real part, and, therefore, only experimental data on the modulus of the dynamic surface elasticity is discussed below. The injection of DNA into the subphase after the formation of the PDAHMAC layer does not lead first to noticeable changes of the surface properties. The surface pressure starts to decrease only after a long induction period, changes from 11 to about 5 mN/m for all the investigated DNA concentrations (7–56 μM) and increases only slightly after that in some cases (Figure 2a). The formation of DNA/polyelectrolyte complexes in the bulk solution is a spontaneous process [4]. The main driving force is considered to be the entropy gain as a result of the release of counterions. If this process occurs in the surface layer, it can lead to a transition of some polyelectrolyte segments from the proximal region of the surface layer into the distal one. As a result, the number of hydrophobic groups in the proximal region of the surface layer decreases a little and the surface pressure starts to decrease leading to a local minimum of the kinetic dependence. The subsequent increase of the surface pressure can be attributed to a transition of some DNA/PDAHMAC complexes from the distal region into the proximal region of the surface layer, and to a slight increase of the local concentration of hydrophobic groups in the latter region. Note that surface pressure never reaches the initial value and the whole spontaneous process results in an increase of the surface tension. This is a relatively rare case in the interface science, and the contribution of the increase of surface tension in the system free energy is presumably compensated by the entropy increase in the bulk phase.

The increase of the surface tension after the induction period is accompanied by an increase of the dynamic surface elasticity but the latter quantity stops to increase earlier than the surface tension and reaches a local maximum of about 32 mN/m (Figure 2b). Although the induction periods of the surface pressure and surface elasticity always coincide, their durations are not reproducible and do not change regularly with DNA concentration. The strong distinctions in the induction period are even observed for solutions with identical concentrations of the components (Appendix A). The insufficient reproducibility of the induction period was already observed for some systems and can be connected with the poorly controlled processes of the nucleation and the subsequent growth of surface aggregates [48]. The surface properties start to change only when the size and/or surface concentration of the aggregates increase and they start to interact. The increase of the dynamic surface elasticity of polyelectrolyte solutions upon DNA addition has been attributed recently to the formation of a rigid network of DNA/polyelectrolyte aggregates at the solution-air interface [38]. The same process presumably results in a strong increase of the dynamic surface elasticity after the induction period in the system under investigation but it becomes more complicated in the latter case and consists of two main steps at least. After a local maximum the dynamic surface elasticity decreases a little and starts to increase strongly again, up to 67 mN/m at a DNA concentration of 14 μM. The time interval of slight changes of the dynamic surface elasticity corresponding to small local maximum and minimum indicates presumably a transition between two surface structures. Unlike the surface pressure, the surface elasticity does not reach a constant value even in 20 h after the DNA injection indicating that a slow process of the surface structure formation almost does not change a local concentration of hydrophobic groups in the proximal region of the surface layer.

### 3.3. DNA Penetration into a PDAHMAC Layer, Low Surface Pressures

A decrease of the initial surface pressure of PDAHMAC layers below the quasiplateau region leads to noticeable changes of the kinetic dependencies of surface properties after the DNA addition into the subphase. The induction period entirely disappears if the initial surface pressure of the PDAHMAC layer is approximately 7 mN/m (Figure 3a,b). The increase of the DNA bulk concentration from 14 to 56 μM leads to acceleration of the changes of surface properties. The minima of both the surface pressure and the dynamic surface elasticity shift towards shorter times when the DNA concentration increase indicating faster changes of the surface structure. Unlike the results in Figure 2a,b, the kinetic curves of the surface tension and surface elasticity are similar in Figure 3a,b and the both quantities decrease just after the DNA injection.

Lowering the surface pressure leads to a more homogeneous PDAHMAC layer without dense aggregates, which hinder the interaction of the polyelectrolyte layer with DNA at higher initial surface pressures. In this case, it follows that the nucleation step of the mixed aggregate formation disappears, and the interaction between the polyelectrolyte and DNA is determined mainly by the DNA diffusion from the bulk phase to the surface, the kinetic dependencies of the surface properties depend clearly on the DNA concentration and the induction period disappears. Note that the pure PDAHMAC layer is less stable at a surface pressure of 7 mN/m, and both the surface pressure and surface elasticity of the layer increase slowly as almost linear functions of time in the course of constant surface oscillations for more than 15 h (Figure 3a,b, black squares). This effect can be connected with slow changes of PDAHMAC conformations in the surface layer leading to the formation of a denser layer.

The further decrease of the initial surface pressure down to 2 mN/m leads to an even faster formation of the mixed DNA/PDAHMAC layer at the total DNA concentration of 28 μM, the surface pressure becomes almost constant in about 4 h after the DNA injection while the dynamic surface elasticity continues to increase slightly a little longer (Appendix A).

### 3.4. Microscopic Morphology of Mixed DNA/Layers

The application of AFM to mixed adsorption layers after their transfer from the liquid surface onto the mica surface by the Langmuir-Schaefer technique allows tracing the main steps of the layer formation. Although the pure PDAHMAC layers are not homogeneous, especially in the phase transition region (Appendix A), they do not contain any threadlike elongated aggregates at all the investigated surface pressures. At the same time, in the mixed layers, some separate and interconnected DNA molecules, presumably together with PDAHMAC molecules, can be observed if the mixed layer is transferred onto the mica surface at low surface pressures close to their minimal values (Figure 4a,b). If the dynamic surface properties reach steady state values prior to transfer, a network of elongated aggregates can be seen on the mica surface (Figure 4c). While the width of the observed elongated aggregates is much higher than the diameter of a double stranded DNA molecule (~2 nm, [49]), their height is close to this value (Appendix A). Note that the width of surface aggregates cannot be resolved with high accuracy because the measured value is comparable with that of the tip curvature of the AFM instrument. Nevertheless, the obtained AFM images show that the observed aggregates consist of a number of DNA molecules. A similar DNA/PDAHMAC network can be also observed on the images corresponding to a lower initial surface pressure of the layer (Figure 4d).

The main distinctions of the DNA/PDAHMAC networks in the given study from the surface networks, which are formed as a result of adsorption of the both components [38], or from DNA/surfactant networks [27], consist in their more regular shape and in a narrower distribution of the cell size in the first case (Figure 4 and Appendix A). Note that the micromorphology of DNA/PDAHMAC network is not always reproducible, and a denser network (Appendix A), as well as separate patches of the network (Appendix A) can be observed occasionally. This is presumably a result of both the disadvantages of the applied transferring method and the nature of the investigated system. For example, the negatively charged DNA molecules can fail to adhere to the similarly charged mica surface and/or the DNA/PDAHMAC network can be broken or compressed during the transfer and subsequent drying.

### 3.5. Ellipsometric Results

Ellipsometry allows tracing the increase of the DNA adsorbed amount in the course of formation of mixed DNA/PDAHMAC layers at the solution—air interface, and fluctuations in the measured values provide insight into lateral inhomogeneities in the surface layer. The changes of the ellipsometric angle ψ with a DNA surface concentration are close to error limits for the system under investigation and are not reported below. Spreading of PDAHMAC onto the surface of buffer solutions changes the ellipsometric angle ∆ only slightly, approximately by 0.590 and 0.740 at surface pressures of 7 and 11 mN/m, respectively, while the DNA adsorption results in a few times larger changes (Figure 5 and Figure 6). The induction periods are clearly seen in Figure 5 at the initial surface pressure 11 mN/m, although their exact durations differ from those in Figure 2a,b probably due to ill-controlled experimental factors influencing the exact rate of aggregate formation. Besides, the difference in the size of measuring cells can also influence the natural convection and thereby the adsorption kinetics. The strong fluctuations of the ellipsometric signal during the induction period show that the layer is not homogeneous on the macroscopic scale, fluidlike and the lateral features moves in and out the reflection spot of the laser beam [44]. Importantly, the fluctuations are only in the positive ∆ direction, which shows that the lateral feature have a thickness of some nm and not much greater. This result supports the AFM result above, where it was indicated that the aggregates comprised a number of entangled DNA molecules rather than any thicker morphology.

The abrupt increase of the ellipsometric angle beyond the period where it was almost constant means that the data from ellipsometry corroborates the assumption discussed above, that a nucleation period is required before the fast growth of DNA/PDAHMAC aggregates. The formation of the complexes of the two polyelectrolytes is obviously hindered by an energetic barrier, which can be connected with the reorientation of some PDAHMAC segments in the surface layer. Only when some nuclei of the aggregates are formed, the aggregation becomes very fast and leads to the DNA incorporation into the PDAHMAC layer (Figure 5). If the surface pressure is below the quasiplateau region of the PDAHMAC compression isotherm, the induction period and energetic barrier disappear and the complexes start to be formed in the surface layer just after the DNA injection (Figure 6). This distinction is presumably caused by a looser PDAHMAC packing in the surface layer in this case. On the contrary, at higher surface pressures the rigid polyelectrolyte aggregates do not allow DNA incorporation in the layer, and the interaction with the synthetic polyelectrolyte requires relatively rare fluctuations of the layer structure. At low surface pressures the complex formation does not lead immediately to the layer immobilization, and it is possible to observe some fluctuations of the ellipsometric signal even in the course of the increase of the DNA adsorbed amount. The higher initial surface pressure, and therefore the higher PDAHMAC surface concentration, result in the higher DNA surface concentrations, and in this case the deviation of the ellipsometric angle from the value for a buffer solution Δ_s_ increases up to 2.10 as compared with 1.60 at the initial surface pressure of 7 mN/m and the bulk DNA concentration of 28 μM (Figure 5 and Figure 6).

An important peculiarity of the obtained results is a small local maximum of the ellipsometric angle ∆ close to the end of the fast increase of this quantity. It is possible to connect this feature with the beginning of the second step of the mixed layer formation. Presumably the first step consists in the fast formation of relatively unordered surface aggregates while the second step results in the formation of more ordered threadlike aggregates (Figure 4 and Appendix A) and a more stable layer when the system free energy is lower. The second step can start with a slight desorption of some DNA segments and therefore a slight decrease of ∆ while the DNA adsorbed amount increases a little again when the system approaches equilibrium (Figure 5 and Figure 6).

### 3.6. IRRAS

Unlike ellipsometry, IRRAS spectra allow separation of the signals from different components in the mixed adsorption layer. For the systems under investigation, the spectral bands in the range 2920–2980 cm^−1^ correspond to the vibration of methylene and methyl groups [50], while the bands in the ranges 1060–1100 cm^−1^ and 1210–1260 cm^−1^ are mainly due to the symmetric and asymmetric stretching of phosphate groups of DNA [22]. The relatively intensive band close to 1090 cm^−1^ gives a possibility to trace the DNA adsorption and incorporation into the PDAHMAC layer (Figure 7a). In about an hour after the DNA injection. it is impossible to observe any indications of DNA adsorption. The intensity of this band starts to increase slightly only in approximately two hours after the injection. The most pronounced increase occurs from 156 to 294 min of the mixed layer formation. At longer times, all the changes are inside error limits. These results agree qualitatively with the kinetic dependencies of other techniques above in Figure 2a and Figure 5. The change of the spectral band with time is slower than that of the ellipsometric angle ∆ (Figure 5) but faster than the change of the dynamic surface elasticity (Figure 2a). The observed distinctions in the rates of surface properties are caused by the influence of uncontrolled experimental factors affecting the exact rate of aggregate formation, as discussed above. At the same time, all the properties almost do not change just after the DNA injection (induction period) at the initial surface pressure 11 mN/m, change faster after this period and finally vary only slightly at the approach to equilibrium.

Approximately the same changes occur in the spectrum range 2920–2980 cm^−1^ (Figure 7b). A slight band at approximately 2955 cm^−1^ for a pure PDAHMAC layer can be attributed to methylene groups of the synthetic polyelectrolyte. The intensity of this band starts to change a little approximately 100 min after the DNA injection, and almost does not change further 300 min after the injection. In spite of the lower intensity of this band, the main steps of its increase are clearly observed and similar to the changes of the bands corresponding to phosphate groups. They are presumably caused by an increase in the surface concentration of methylene groups due to adsorption of the nucleic acid and indicate that DNA can be the main component of the mixed layer. Note that the detected asymmetric methylene and methyl bands are very weak and the RA corresponding to band maxima exceeds experimental errors only slightly.

## 4. Discussion

The novelty of the system under investigation lies in the ability to tune the solubility of a synthetic polyelectrolyte by changing the ionic strength of the subphase to examine interactions of either its adsorbed or spread layers with a natural polyelectrolyte. This feature strongly distinguishes this system from the usually studied mixed layers of phospholipids and DNA where the lipid solubility is negligible as compared with that of PDAHMAC [3,12,13,15,16,17,19,20,21,22,23,26,32,33,34]. The DNA–lipid interactions are mainly considered to be of electrostatic nature, and DNA is attracted to the layer of positively charged groups of the lipid and can penetrate it [20,21,22,34,51]. These interactions can result in surface conformational transitions and/or the formation of surface aggregates [22,26,34]. The aggregate shape depends on the system but usually is similar to that of lipid surface aggregates. Some ribbons were observed in case of DNA penetration into model negatively charged lipid monolayers on an aqueous subphase containing DNA and CaCl_2_ and at high surface pressures only (~40 mN/m) [26]. The ribbonlike aggregates are also formed if DNA interacts with insoluble monolayer of Gemini surfactants [24].

Studies of the DNA/surfactant aggregate formation in the surface layer as a result of simultaneous adsorption of the both components from the aqueous phase are much rarer [14,27]. Recently a network of ribbonlike aggregates has been observed at the surface of mixed DNA solutions with cetyltrimethylammonium bromide [27]. A network that is similar, although a little more regular, was also discovered at the surface of mixed DNA/PDAHMAC solutions in a narrow range of concentration ratios [38]. It was assumed that the network is formed as a result of adsorption of mixed aggregates from the bulk aqueous phase. The results of the present study do not corroborate this assumption. The formation of mixed DNA/PDAHMAC aggregates in the systems investigated in the present work occurs directly in the surface layer where the PDAHMAC molecules are localized.

Another important conclusion from the present work is that the rate of interaction of the two polyelectrolytes depends strongly on the packing of PDAHMAC molecules in the surface layer. If these molecules are close packed in the surface layer at surface pressures higher than the beginning of the quasiplateau region, the rate of interaction is very low and presumably determined by fluctuations of the surface structure. It is possible to assume that a local destruction of the PDAHMAC layer is necessary to form a DNA/PDAHMAC complex in the surface layer. The hexyl groups in the PDAHMAC layer are directed to the air phase and if the layer is close packed, there are some steric obstacles for phosphate groups of DNA to approach a nitrogen atom of the synthetic polyelectrolyte (Figure 8a). Although the release of counterions of DNA and PDAHMAC is presumably the driving force of the complex formation, the counterions are not shown in the figure for the simplicity. If the concentration of local distortions of the PDAHMAC layer exceeds a certain critical value, the rate of complex formation increases strongly in cases after an induction period, and one can then observe fast changes of the surface properties, as demonstrated in the present work using complementary surface techniques. At surface pressures lower than the quasiplateau region, the PDAHMAC layer is fluidlike and is subject to stronger fluctuations (Figure 8b). In this case the energetic barrier for the complex formation decreases, and the rate of the DNA/PDAHMAC interactions increases strongly leading to disappearance of a noticeable induction period of the kinetic dependences of surface properties.

The formation of the first DNA/PDAHMAC complexes presumably does not lead to a minimum of the system free energy and the layer morphology continues to change further. The difficulties of layer transfer to the mica surface in the Langmuir−Schaefer method, the layer heterogeneity and relatively fast changes of the morphology do not allow tracing this process by AFM and give a possibility to observe only the final state of the mixed layer formation. The obtained results indicate that close to equilibrium the layer consists of threadlike or ribbonlike fibrous aggregates forming a network with the mean cell size in the range 100–200 nm. The height of aggregates is in the range 1.5–2.5 nm and close to the diameter of the DNA double helix. The width of the aggregates is much larger than these values indicating that they consist of a few DNA molecules presumably with some PDAHMAC chains. The main distinction from the DNA/PDAHMAC network observed previously at the surface of mixed solutions consists in a more narrow distribution of the cell size if the synthetic polyelectrolyte is spread onto the aqueous surface [38]. The observed networks of the two polyelectrolytes resemble a little those in some other similar systems [24,26], but appear to be more regular. It may be noted that we have never observed compact polyplexes, which are characteristic for bulk phases of the mixed solutions containing DNA, at the surface of aqueous phases [4].

Research into the formation of 2D nanostructures attracts a lot of attention [1,2,9,10]. While many of the systems investigated involve lipid monolayers [11,12,13,15,16,17,19,20,21,22,23,26,34,51], the present work has adopted a novel approach to confine a spread polyelectrolyte film on a subphase of elevated ionic strength to gain insight into the mechanism of DNA interactions that determine surface aggregate formation. The results have shown that various surface properties including the dynamic surface elasticity, surface tension and surface excess exhibit a long induction period only if the initial surface excess of the surface film is in a quasiplateau region and not if the initial surface excess is lower. This information allows us to propose a mechanism of interaction. A future direction of the work could be to apply a technique such as neutron reflectometry to gain supporting insight into the interfacial structure including the volume fraction and thickness of the layers [18,52], although clearly this remains outside the scope of the present work. Given the use of DNA/polymer systems in research on gene delivery [53] and electronic devices [54], the new insight into the mechanism of formation of their nanostructures at interfaces provided by the present work can have a variety of future implications.

## 5. Conclusions

Recent studies have shown that a network of fibrous aggregates can be formed at the surface of mixed solutions of DNA and surfactants or amphiphilic polymers. In the present work the spreading of a hydrophobic cationic polyelectrolyte (PDAHMAC) on an aqueous subphase with elevated ionic strength and the subsequent DNA injection into the subphase allowed localization of the interactions between the two substances, which lead to the formation of the regular network. The formation of fibrous aggregates in the surface layer can result in non-monotonic kinetic dependences of surface properties with a few local maxima and minima. It has been shown that the rate of network formation depends strongly on the packing of PDAHMAC molecules in the surface layer, and thereby can be tuned by changes of the surface pressure. At the same time, the micromorphology of the mixed PDAHMAC/DNA does not depend noticeably on the initial surface pressure of the layer. The formation of compact polyplexes, which are characteristic for bulk phases of the mixed solutions with DNA, was not observed at the solution—air interface.

## Figures and Tables

**Figure 1 polymers-13-02820-f001:**
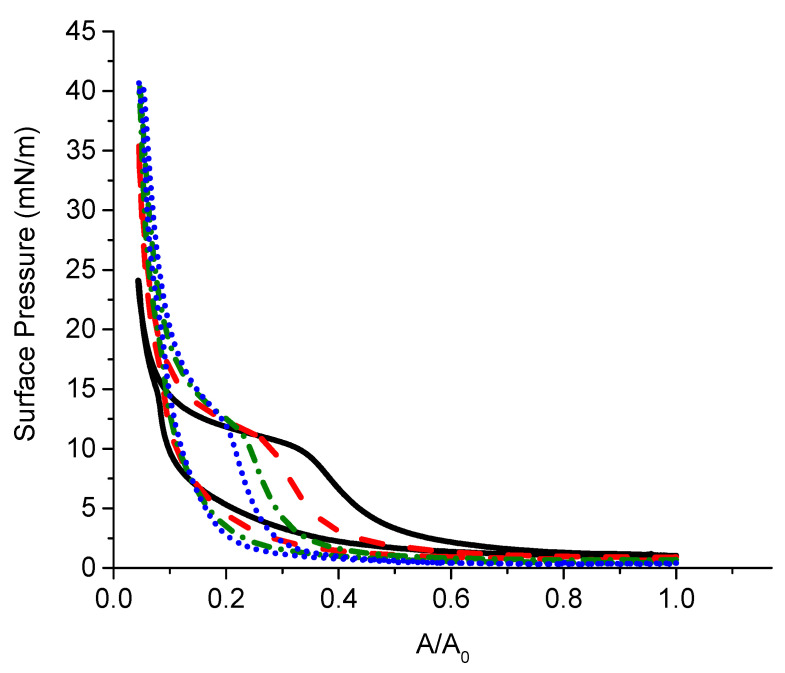
Surface pressure vs. relative surface area A/A0 of a PDAHMAC layer spread onto buffer solution. Lines correspond to consecutive compression-expansion cycles: first cycle (black), second cycle (dashed red), third cycle (dash-dotted green) and fourth cycle (dotted blue).

**Figure 2 polymers-13-02820-f002:**
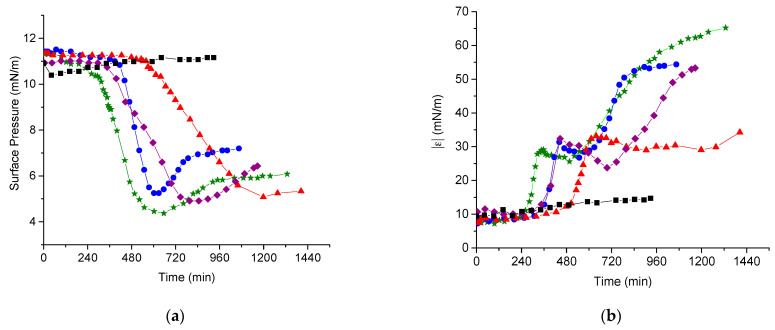
Kinetic dependences of the surface pressure (**a**) and dynamic elasticity (**b**) of PDAHMAC layers after DNA injection into subphase. The DNA concentrations in the subphase are 0 (black squares), 7 (red triangles), 14 (green stars), 28 (blue circles) and 56 μM (purple diamonds). The initial surface pressure of the layer is 11 mN/m. Lines are guides for an eye.

**Figure 3 polymers-13-02820-f003:**
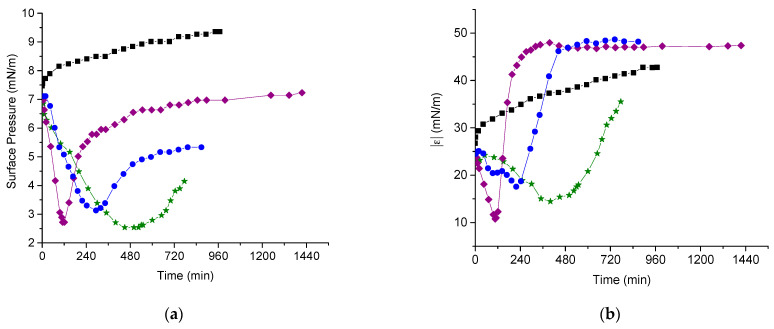
Kinetic dependences of the surface pressure (**a**) and the dynamic elasticity (**b**) of a PDAHMAC layer after DNA injection into subphase. The DNA concentrations in the subphase are 0 (black squares), 14 (green stars), 28 (blue circles) and 56 μM (purple diamonds). The initial surface pressure of the layer is 7 mN/m. Lines are guides for an eye.

**Figure 4 polymers-13-02820-f004:**
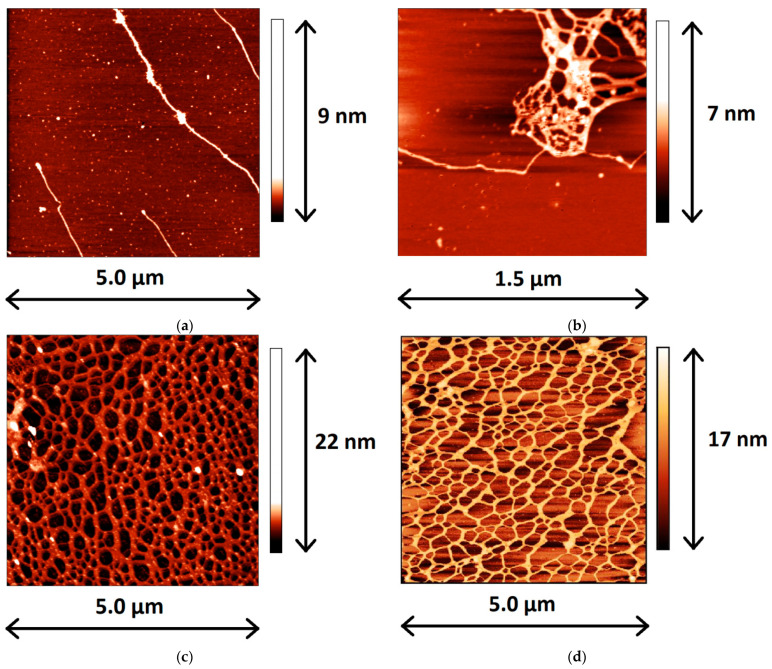
AFM images of spread PDAHMAC layers transferred onto a mica surface in 600 (**a**,**b**) and 1320 min (**c**) after DNA injection into subphase. The initial surface pressure of the layer is 11 mN/m and the total DNA concentration is 14 μM. Image (**d**) corresponds to 840 min after a DNA injection into subphase at the initial surface pressure of 7 mN/m and a total DNA concentration of 28 μM.

**Figure 5 polymers-13-02820-f005:**
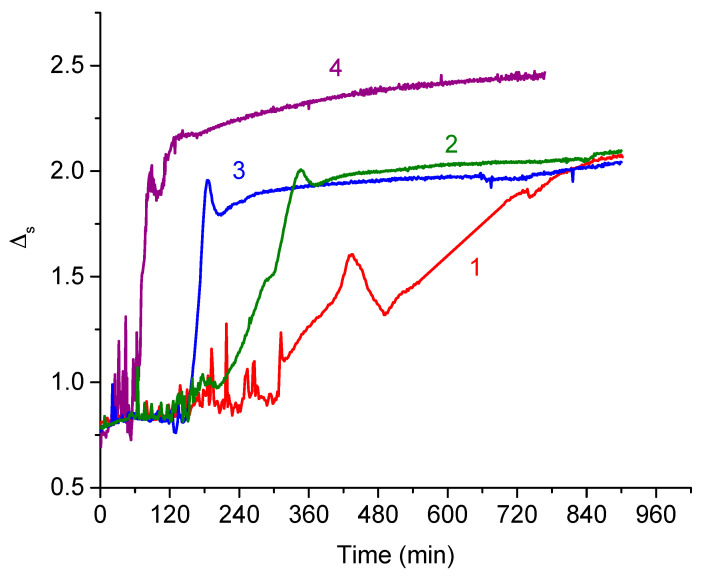
Kinetic dependencies of the ellipsometric angle Δ_s_ of the PDAHMAC layer after DNA injection into subphase. The total DNA concentrations are 7 (1), 14 (2), 28 (3) and 56 (4) μM. The initial surface pressure the PDAHMAC layer is 11 mN/m.

**Figure 6 polymers-13-02820-f006:**
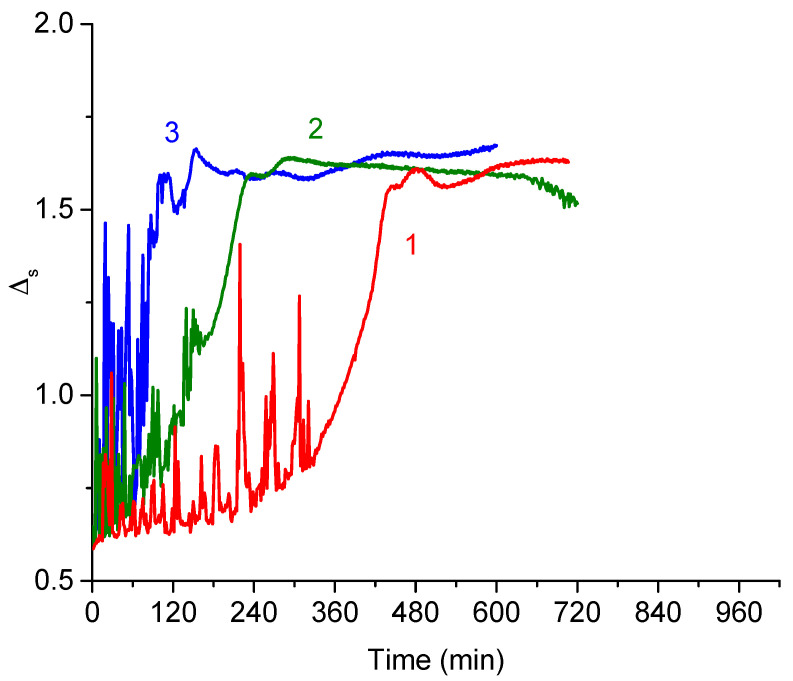
Kinetic dependencies of the ellipsometric angle Δ_s_ of the PDAHMAC layer after DNA injection into subphase. The total DNA concentrations are 7 (1), 14 (2) and 28 (3) μM. The initial surface pressure of the PDAHMAC layer is 7 mN/m.

**Figure 7 polymers-13-02820-f007:**
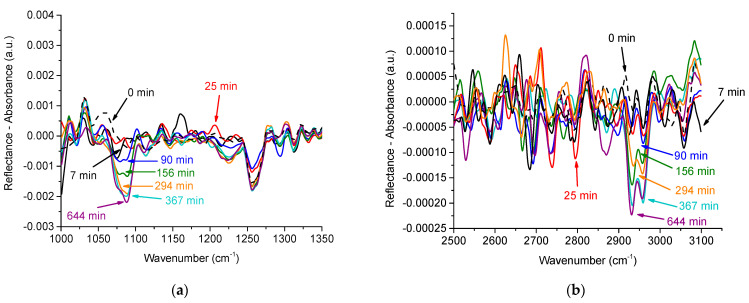
IRRAS spectra 1000–1350 cm^−1^ (**a**) and 2500–3100 cm^−1^ (**b**) of a PDAHMAC layer in different times after the DNA injection into subphase. The total DNA concentration is 56 μM and the initial surface pressure of the PDAHMAC monolayer is 11 mN/m. Dashed black line corresponds to the PDAHMAC monolayer before the DNA injection.

**Figure 8 polymers-13-02820-f008:**
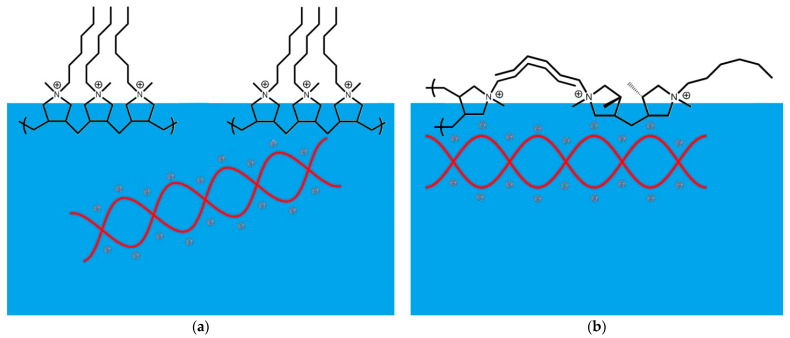
A scheme of inferred DNA interactions with a PDAHMAC layer at surface pressures in the region of a two-dimensional phase transition (**a**) and below this region (**b**) prior to aggregate formation.

## Data Availability

Data is contained within this article.

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
