# Peer review of "DNA Interaction with a Polyelectrolyte Monolayer at Solution—Air Interface"

_polymers, 2021, doi:10.3390/polym13162820_

Round 1
Reviewer 1 Report
The manuscript reported to utilize a new approach to confine the hydrophobic cationic polyelectrolyte (poly(N,N-diallyl-N-hexyl-N-methylammonium) chloride) at the solution – air interface by spreading it on a subphase with elevated ionic strength. Several techniques were applied to gain insight into the main steps of the adsorption layer formation. Overall the topic is interesting and the manuscript is well written. It is suggested to be accepted after minor revision.
- For Figure 8, chloride should be not omitted. The authors should provide some characterization data for poly(N,N-diallyl-N-hexyl-N-methylammonium) chloride.
- It is suggested to give the specific preparation process for poly(N,N-diallyl-N-hexyl-N-methylammonium) chloride, instead of just reference citing.
- What is the potential application of this research?
4. It is mentioned in the manuscript that the present results provide new perspective on nanostructure formation involving nucleic acids building blocks. The authors should elaborate more on this point.
Author Response
Reviewer 1
The manuscript reported to utilize a new approach to confine the hydrophobic cationic polyelectrolyte (poly(N,N-diallyl-N-hexyl-N-methylammonium) chloride) at the solution – air interface by spreading it on a subphase with elevated ionic strength. Several techniques were applied to gain insight into the main steps of the adsorption layer formation. Overall the topic is interesting and the manuscript is well written. It is suggested to be accepted after minor revision.
Authors’ reply:
We are grateful to the reviewer for finding our manuscript interesting and deserving publication.
- For Figure 8, chloride should be not omitted. The authors should provide some characterization data for poly(N,N-diallyl-N-hexyl-N-methylammonium) chloride.
Authors’ reply:
The main idea of Figure 8 is to illustrate the steric hindrances, which appear when the PDAHMAC monolayer is compressed and affect its interaction with DNA. In order to sharpen this point and simplify the figure, all other species present in the solution are omitted. To make this idea clearer the following sentence has been included in the new version of the manuscript (p. 14): “Although the release of counterions of DNA and PDAHMAC is presumably the driving force of the complex formation, the counterions are not shown in the figure for the simplicity”.
The NMR data (1Đť, D2O, 400 MHz) for poly(N,N-diallyl-N-hexyl-N-methylammonium) chloride are added now to the Materials and Methods section of the new version of the manuscript (p.3): “NMR 1Đť (D2O, 400 MHz) δ, ppm: 3.95 (br. m, 2H, Pyrr-2,5 pseudo equatorial), 3.47-3.38 (br. m, 4H, Pyrr-2,5 pseudo axial, Hex(1)) 3.25 & 3.15 (s, 3H, CH3N), 2.77 (br. m 1.6H, Pyrr-2,3 cis), 2.33 (br. m 0.4H, Pyrr-2,3 trans), 1.98-1.22 (br. m, 12H, CH2-Pyrr, Hex(2-5)), 0.98 (br. t, 3H, Hex(6)).”
- It is suggested to give the specific preparation process for poly(N,N-diallyl-N-hexyl-N-methylammonium) chloride, instead of just reference citing.
Authors’ reply:
The preparation process for poly(N,N-diallyl-N-hexyl-N-methylammonium) chloride is described now in the Materials and Methods section of the new version of the manuscript (ps. 2-3): “The monomer (2.225 g, 9.6 mmol), 4,4′-azobis(4-cyanovaleric acid) (84 mg, 0.3 mmol) and deionized water (0.690 g) were vortexed in a glass ampoule until dissolution, the mixture was subjected to three freeze-pump-thaw cycles in order to remove oxygen, sealed and heated at 70°C for 13 hours. The highly viscous product was dissolved in water and dialyzed against deionized water for three days using Spectra/Por MWCO 6-8 000. The solution was concentrated in vacuum, frozen and lyophilized. White sponge-like product was dried in vacuum over phosphorous pentoxide, yield: 75.1%, intrinsic viscosity: 0.30 dL/g (1 M NaCl, 25 °C)”.
- What is the potential application of this research?
Authors’ reply:
The new references related to the applied significance of the obtained results are added to the new version of the manuscript and some potential applications are described now on page 2: “The subsequent transfer to a solid substrate by Langmuir–Blodgett technique after such binding can be employed to fabricate DNA nanostructures of different morphology [28], which find their use in biosensing applications [29,30]. However, little is known about the mechanism of complex DNA layer formation, while this information may be vital for the production of devices with the desired properties”.
- It is mentioned in the manuscript that the present results provide new perspective on nanostructure formation involving nucleic acids building blocks. The authors should elaborate more on this point.
Authors’ reply:
The obtained results give an example of the formation of regular nanostructures from double stranded DNA molecules. Nowadays the DNA-based nanostructures are usually formed with single stranded molecules. In this case, the building of nanostructures requires a few steps, which can be rather complicated. In our approach the nanostructures are formed spontaneously at the liquid – gas interface and can be easily transferred onto a solid support. These points are considered in the introductory section.
Reviewer 2 Report
The manuscript ‘DNA Interaction with a Polyelectrolyte Monolayer at Solution – Air Interface’ presented an interesting topic. Many techniques including rheology, microscopy, ellipsometry, and spectroscopy are used to provide information on the interaction. It is well designed, carried out and written. I agree to accept the manuscript at current version.
Author Response
The manuscript ‘DNA Interaction with a Polyelectrolyte Monolayer at Solution – Air Interface’ presented an interesting topic. Many techniques including rheology, microscopy, ellipsometry, and spectroscopy are used to provide information on the interaction. It is well designed, carried out and written. I agree to accept the manuscript at current version.
Authors’ reply:
We are grateful to the reviewer for the appreciation of our results and the recommendation to publish our manuscript.